# Micro, Small, and Medium Enterprises' Readiness for Digital Transformation in Indonesia

## Lina Anatan * and Nur

Faculty of Business, Maranatha Christian University, Bandung 40164, Jawa Barat, Indonesia
* Correspondence: lina.anatan@eco.maranatha.edu

**Abstract:** The Fourth Industrial Revolution (IR4) and the COVID-19 pandemic have become triggers for micro, small, and medium enterprises (MSMEs) to conduct digital transformation even though there are many problems that need to be resolved, particularly those related to the readiness of MSMEs in facing digitalization. This study aims to investigate Indonesian MSMEs and identify problems and types of knowledge transfer activities. By involving 101 MSMEs selected using convenience sampling and collected through an online survey, the hypotheses testing shows that the perception of higher drivers for IR4 promoting IR4 readiness is supported, while the perception of higher barriers to IR4 decreasing IR4 readiness is not supported. The problems faced by MSMEs in Indonesia are related to financial, human resources, marketing, operational, administrative, and organizational management. To solve these problems and enhance the readiness for digitalization, knowledge transfer activities from universities to MSMEs are needed. This study provides a theoretical contribution to the strategic management literature to fill the lack of studies on MSMEs' e-readiness in developing countries and a practical contribution to assist decision-makers in formulating strategies to support MSMEs in facing IR4 and solving internal problems through knowledge transfer activities.

**Keywords:** MSME readiness; IR4; digital transformation; knowledge transfer

## 1. Introduction

Micro, Small, and Medium Enterprises (MSMEs) are the pillars of the Indonesian economy that significantly contribute to the achievement of Indonesia's Gross Domestic Product (GDP). In 2015, MSMEs contributed 99.98% of business units with a contribution rate of 57% to Indonesia's GDP and absorbed more than 97% of the domestic workforce (Ashariyadi 2016). In 2020, the contribution of MSMEs to Indonesia's GDP increased to 61%, with the absorption rate of domestic workers reaching 97% (kominfo.go.id (accessed on 23 December 2022)).

The latest data show that, until 2022, MSMEs contributed to 61.1% of Indonesia's GDP, and the remaining 38.9% was contributed by only 5,550 large businesses, or 0.01% of the total business actors (Limanseto 2022). Of the 61.1% of MSMEs' contribution to Indonesia's GDP, it is dominated by micro-entrepreneurs, amounting to 98.68% with a workforce absorption capacity of around 89%. The magnitude of the contribution of MSMEs to the Indonesian economy means that Indonesia has a potentially strong national economic base due to many MSMEs, especially those on a micro scale, with quite a large absorption capacity.

Despite having an important and significant role in the national economy, MSMEs in Indonesia are faced with many internal and external problems to be resolved. The internal problems include, firstly, institutional and human resource problems, namely a lack of knowledge, competence, and human resource capacity; secondly, production and marketing problems, namely the limitations of MSMEs' ability to innovate products and packaging, utilize digital technology in marketing, and a lack of understanding regarding the importance of brands and customer relationship management; and thirdly, the limitations of intellectual property and a low level of digital literacy. These problems become



worse due to the lack of MSMEs' understanding of the importance of cooperation with external parties such as universities, banks, and other institutions.

The external problems of Indonesian MSMEs can be classified into financial problems, bureaucratic services, and infrastructure. Financial problems are related to the difficulty of gaining access to funding since there are many MSMEs that are unfamiliar with financial institutions. The low level of MSMEs' digital literacy also becomes a trigger, since many financial technology services available are unoptimized and unused. The lack of understanding of the importance of legal aspects and the lack of optimal infrastructure support for digital development are the other external problems that should be considered by MSMEs, especially in the IR4 era.

Internal problems related to limited resource aspects (for example, lack of knowledge related to organizational management, low competence of human resources, low knowledge of digital marketing technology adoption, organizational financial management, the importance of Standard Operational Procedures (SOP), other decisions in operational functional areas, and other problems) can be overcome through knowledge transfer from universities to MSMEs through partnership mechanisms. Through partnerships, MSMEs may obtain various advantages, such as access to resources, knowledge, and skills that they do not have. The need for access to resources, core competencies, innovative skills, and specific knowledge are the main motivators for the formation of partnerships between organizations (Hamel et al. 1989; Kogut and Zander 1992).

The problem of limited resources and the low digital literacy of Indonesian MSMEs is an important problem that must be resolved immediately to increase the productivity and performance of MSMEs in the IR4 era. Several strategies have been implemented by the Government of Indonesia to address the limited resources and strengthen the digital literacy of MSMEs in Indonesia, one of which is to involve external parties such as universities to increase knowledge transfer activities from universities to MSMEs so that the problems can be overcome or at least minimized.

Knowledge transfer activities focus on developing human resource capacity (through EDUKUKM, the SPARC Campus Webinar Series, and the MSME Digital Hero Program), community development, and local applications (such as Gotong Royong Market and Indonesia Creative Store), MSME digitalization programs (such as MSME Foster Brothers, Digital Catalog, and IMOOJI), and marketing or promotion support (through some approaches such as product reviews by influencers, SPARC Trade, KUKMHUB/e-commerce, and product promotion by PLUT KUMKM).

EDUKUKM is a form of MSMEs e-Learning media that is a manifestation of the Government's contribution, particularly the Ministry of Cooperatives and SMEs (Small and Medium Enterprises), in order to provide provisions and skills to MSMEs, especially during the COVID-19 pandemic, so that they can continue to improve competence (https://edu.kemenkopukm.go.id/ (accessed on 23 December 2022)). The SPARC Campus webinar series involves the Government, academics, community/society, media, professionals, influencers, and other private sectors. It is provided by SMESCO, an institution under the Indonesian Ministry of Cooperatives and MSMEs, as a method of preparing MSMEs to face the challenges of the business environment (smesta.kemenkopukm.go.id (accessed on 23 December 2022)). The Digital Hero Program is a collaboration program between Jagoan Hosting and the Ministry of Communication and Information to encourage MSMEs that have not gone online yet to re-package and re-brand their products so they can compete in the international market (Anatan and Ellitan 2022).

The Gotong-Royong Market was developed to provide solutions to MSMEs' problems during the pandemic and to promote #BanggaBuatanIndonesia. It is a collaborative program between KlikLunas and the Ministry of Cooperatives and MSMEs. Meanwhile, Indonesia Creative Store is a creative product catalog developed by the Indonesia Creative Cities Network (ICCN) and contains selected products from all MSME networks in Indonesia.

The MSME Foster Brother Program aims to expand the MSME product market and provide training to increase human resource capacity. It is a collaboration between the Ministry of Cooperatives and MSMEs and BliBli. Digital Catalog is a website developed to promote MSME products via the Internet, such as the digital catalog of MSMEs in the city of Semarang, which contains all types of products produced by all MSME business actors in the city of Semarang (Anatan and Ellitan 2022), while Imooji is a digital platform used to communicate through making interactive digital brochures, product catalogs, event invitations, promotion of goods or services, greeting cards, and so on (imooji.com, accessed on 23 December 2022).

These strategies were developed to prepare MSMEs to respond to IR4. IR4 provides challenges and opportunities for Indonesian MSMEs to be able to compete at the international level of competition. To succeed, Indonesian MSMEs should have a high level of readiness to overcome challenges and respond to existing opportunities. According to the data published by the Ministry of Information and Communication, the total number of MSMEs in Indonesia in 2020 reached 64.2 billion; however, of the total MSMEs, only 17.1% were technology-literate and utilized e-commerce as an online sales channel in early 2020. The digital literacy rate of MSMEs in Indonesia is still quite low and 81% of MSMEs in Indonesia have not been touched by digitalization (Natalia 2021).

Sugihartati (2023) stated that based on data from the Central Statistics Bureau, in June 2021, out of 8.2 million business units surveyed, only 29% of MSMEs had used e-commerce. In August 2022, data on MSMEs that had entered the digital ecosystem reached 20.24 million. In 2023, the Government of Indonesia aims to target 30 million Indonesian MSMEs to enter the digital ecosystem using marketplaces or digital platforms.

Studies on the readiness of Indonesian MSMEs to face digital-based business transformation because of IR4 and the low-touch policies during the COVID-19 Pandemic, better known as e-readiness, have been conducted several times. Panjaitan et al. (2021) conducted a study that aimed to explore the technological readiness of MSMEs and their ability to compete digitally by involving 170 respondents. Data were collected using a questionnaire and analyzed using SEM-PLS. The results of data processing show that digital value mediates the relationship between technological readiness and digital capability. In addition, changes in consumer tastes and habits for digital transactions have also been proven to strengthen the relationship between technological readiness and digital competitive capabilities.

Rafiah et al. (2022) conducted a study using descriptive statistics to test MSME's readiness level for digitalization by involving 113 MSMEs in West Java and 5 variables; namely, people, process, strategy, technology, and integration. The results of the study show that the level of MSME readiness for digitalization is quite low, and of the five indicators used, people have the highest level of readiness. MSMEs are still in the stage of taking the initial steps to determine the best strategy to prepare for digitalization. Meanwhile, the results of a study conducted by Harmawan (2022) show that MSMEs in Banyuwangi who have utilized digital marketing through Banyuwangi-mall.com have a higher level of readiness for technology adoption in terms of strategy and management.

The differences in study results related to the readiness of MSMEs to conduct digital-based transformation motivated researchers to test the level of readiness for digitalization of MSMEs or e-readiness based on the aspects of e-readiness drivers and barriers. Previous studies on MSME e-readiness in Indonesia emphasized the readiness of MSMEs in adopting e-commerce and promotion using electronic media; however, e-readiness based on drivers and barriers has not been implemented in Indonesia. Drivers and barriers to e-readiness are considered important to determine the supporting and inhibiting factors for MSMEs' readiness to digitalize so that the proper strategy can be formulated based on these two factors (Stentoft et al. 2019).

A study conducted by Stentoft et al. (2019) involved 308 MSMEs to investigate the MSMEs' readiness for digitalization and their actual practices related to digitization decisions. The results of the study show that the perceived drivers of IR4 have an impact on increasing the level of readiness of IR4, which in turn has an impact on increasing the

practice of IR4. It also found that existing barriers would reduce the level of IR4 readiness, however, would not have a significant impact on IR4 practices.

A study conducted by Turkes et al. (2019) aimed to identify the opinions and perceptions of MSME managers in Romania regarding the drivers and barriers to implementing IR4 technology for business development purposes. By involving 176 MSME managers in Romania, it was found that Romania was in a full transition from the Second Industrial Revolution (IR2) to IR4, where the level of knowledge regarding IR4 technology was quite high and the desire of MSMEs in Romania to implement IR4 technology was also quite high.

This research was conducted to determine Indonesian MSMEs' readiness for IR4 using the studies of Stentoft et al. (2019) and Turkes et al. (2019) as references to develop variable measurement instruments. In this study, respondents were also asked to identify the problems they faced related to readiness for digitalization. Considering the problem of limited resources and the relatively low level of digital literacy faced by MSMEs, and to optimize the role of universities as knowledge producers in dealing with MSME problems, this study also identified knowledge transfer activities from universities to MSMEs required based on the MSME perspective.

According to the previous discussion, the objectives of this study can be identified as follows: (1) to provide empirical evidence that the higher the perception of drivers for promoting IR4, the higher the level of IR4 readiness; (2) to provide empirical evidence that the higher the perception of the barriers to promoting IR4, the lower the level of IR4 readiness; (3) to identify problems faced by Indonesian MSMEs; and (4) to identify knowledge transfer activities from universities required by MSMEs.

The urgency of this research lies in realizing the contribution of universities to the development of knowledge, especially in the field of strategic management. Studies on MSME readiness to respond to the IR4 era have been mostly conducted in the manufacturing industry and are still limited to MSMEs and have led to the increase of research gaps on the related issues (Stentoft et al. 2019). This research is expected to fill the existing gaps, not only to find the result of the identification of the existing problems but also to provide a significant contribution to provide insight on appropriate strategies and policies to improve MSMEs' performance and competitiveness in the IR4 era.

## 2. Theoretical Background and Hypothesis Development

This section will discuss the theoretical review and hypothesis development. The discussion begins with the definition of MSMEs referred to in this study, namely referring to Law No. 8 of 2008 and the MSME digitalization strategy in Indonesia. The following discussion focuses on transferring knowledge from universities to MSMEs and developing hypotheses about the drivers and barriers of MSME e-readiness.

### 2.1. A Brief Review on MSME Digitalization Strategy

The definition of MSMEs regulated in Law No. 8 of 2008 is classified based on asset and turnover criteria. Given the importance of the role of MSMEs in supporting the Indonesian economy, the Government is concerned with accelerating the process of digitizing MSMEs, since this digitization strategy is vital to increasing MSMEs' competitiveness in the IR4 era. On the other hand, since the COVID-19 pandemic occurred in Indonesia at the beginning of March 2020, various restrictions on social activities through large-scale social restrictions have had a very significant impact on reducing MSME turnover, and several MSMEs were even forced to end their operational activities temporarily or permanently. The closure of the businesses was due to the drastic decline in MSME income while operating costs had to be paid.

The study conducted by Baldwin and di Mauro (2020) shows that the various restrictions applied did not only have an impact on the demand side, which ultimately affects MSMEs' income but also on the supply side. The impact on the demand side is related to changes in consumption style and consumer purchasing power. For instance, changes in the

learning and work processes which were originally face-to-face changed to utilizing online platforms such as MS Teams, Zoom Meetings, Google Meet, etc., equally for consumption and shopping activities, as well as entertainment and play. Before the pandemic, window-shopping activities, watching movies in cinemas, and other entertainment activities were carried out in public. However, with various existing restrictions, shopping, consumption, and entertainment activities were carried out online; there was an increase in viewing activities through Netflix as well as requests for subscriptions to Cable TV or YouTube (Anatan and Ellitan 2022).

On the supply side, the pandemic has had an impact on limited raw materials and the mobility of human resources, which has become a serious challenge for MSMEs in Indonesia. So far, MSMEs in Indonesia have always relied on conventional face-to-face transactions or physical transactions and used cash to do business. Another problem relates to the types of MSME products and services that are consumed directly and are not fixed or depend on business conditions. Regarding the profile of MSME business actors in Indonesia, especially those on a micro scale, the majority still have low digital literacy skills and a low level of education (Anatan and Ellitan 2022).

These impacts triggered a disruption in the linear movement pattern of the world, creating a new order pattern that relied on online activities (Reardon et al. 2020). Digitizing businesses became the best solution for MSMEs to survive in the digital-based transformation era due to IR4 and the COVID-19 pandemic. This pandemic has become a kind of "blessing in disguise", since the need for MSMEs to be digitized cannot be delayed any longer. Changing or dying is the choice faced by MSME actors.

Several studies conducted show that digitalization has a significant impact on financial performance (Indriastuti and Kartika 2022; Gunawan and Somantri 2023), revitalization (Putri and Asyari 2023), and the economy in general (Nata et al. 2022). Indriastuti and Kartika (2022) conducted a study involving 282 MSME respondents in Central Java to examine the impact of MSME digitization on MSME financial performance. The study results show that digitization improves financial performance.

Gunawan and Somantri (2023) conducted a study to examine the effect of the digital economy on the financial inclusion of MSMEs with technology adoption as a variable that mediates the impact of the digital economy on financial inclusion. The study results show that the digital economy and technological adaptation have a significant impact on financial inclusion. This means that the digital economy is an indicator of economic growth and development in the future which is marked by rapid developments in business transactions.

The study conducted by Putri and Asyari (2023) aimed to examine the impact of digital transformation on MSME revitalization during the pandemic by involving 278 MSME respondents in Bukittinggi. The results show that digital transformation had a significantly positive effect on MSME revitalization. This finding shows the importance of digital transformation in the revitalization of MSMEs, especially during the pandemic. Researchers also recommend that digital literacy needs to be increased to support digital transformation.

Nata et al. (2022) conducted a study on the effect of digitizing MSMEs on the economy. According to them, by digitizing, MSMEs might overcome various problems faced by creating new business models that are more relevant and adaptive to changes in the business environment. For instance, MSMEs adopting digital marketing will make it easier to expand market share so that income can be increased.

MSME digitization strategies in Indonesia are developed to change the MSME actors' behavior in marketing and selling their business products using digital technology. To face the challenges of IR4, Fajria (2020) suggests that, conceptually, MSME 4.0 is an MSME that can adapt digital technology and utilize digital technology to maintain business in a dynamic digital technology ecosystem.

Practically, MSMEs are encouraged to take advantage of technology, such as WhatsApp, social media, Google Business, and e-commerce platforms. The use of digital technology is essential to support MSME marketing activities and provide benefits in expanding

market share. For instance, MSMEs with offline stores will only reach market share around the store location; however, by developing online stores and online marketing, MSMEs can reach buyers outside the region, city, province, and even abroad.

The process of accelerating Indonesia's MSME digitalization strategy is conducted at the national level and receives complete support from local governments. For instance, the Kendal Regency Government has published a pocketbook that contains guidelines for implementing and accelerating the digitization process for Kendal Regency MSME (Ganinduto 2021). In the guidebook, the regional government explains the importance of digitizing MSME, including increasing market share and income, increasing the workforce absorbed in the MSME sector, reducing unemployment, and as a medium for sharing inspiration and information among MSME actors. The concept of developing MSME through the MSME digitization strategy focuses on four aspects: productivity access, market access, business partners, and promotion.

Productivity aspects include capital (through banking, people's business credit, and corporate social responsibility), technology (i.e., production equipment and product packaging), legal aspects of licensing, and creative hubs. Market access focuses on developing digital platforms, e-commerce, supermarkets, and minimarkets, as well as gift shops. To support the success of MSME market access, they receive digital marketing training from the industry office and MSME cooperatives. Business partners emphasize increasing MSME cooperation with local governments, central government, e-commerce, and private companies. In addition, involvement in business associations and cooperation between countries also require special attention. The promotion aspect focuses on the use of social media, websites, and collaboration with influencers. In addition, MSMEs are also encouraged to be involved in events such as festivals or expos to promote their products.

To succeed in the digital transformation of MSMEs, products are the most important aspect to be considered. Digital MSME products must be of high quality and available on various digital platforms, be professional, have obvious and attractive product descriptions, and have attractive packaging. In addition, to increase customer satisfaction and loyalty, MSMEs must be able to manage reviews and feedback well and provide quick responses to customer orders. To increase market share, the use of digital platforms is necessary, including social media such as Instagram, Facebook, TikTok, and YouTube. Other digital platforms could also be used, such as the Gojek and Grab super-apps, Google platforms such as Google Business and Maps, and e-marketplaces such as Lazada, Tokopedia, Shopee, or Bukalapak. MSME actors might also gain an advantage from influencers' services to introduce and market their products so that sales and market share could be increased.

### 2.2. Knowledge Transfer from University to MSMEs

MSMEs in Indonesia face various problems, both internal and external, related to limited resources, limited knowledge, and expertise in various matters of business management. To overcome these problems, MSMEs may collaborate through a partnership with other parties, such as universities, so that knowledge transfer activities from universities to MSMEs can be conducted. In this partnership scheme, the university acts as a transferor, and the MSME as a transferee, and the mechanism and type of cooperation can be conducted via various approaches depending on the MSME's needs.

Several forms of knowledge transfer from universities can be made according to MSMEs' needs, including technology development, human resources development, and access to experts. Firstly, regarding the development of technology for commercialization purposes, knowledge transfer from universities can be conducted by providing knowledge through workshops, training, and technical guidance on the use of digital marketing in the MSME business. Digital marketing involves the use of the internet and interactive technology such as blogs, websites, email, AdWords, and social networks to carry out marketing and promotional activities, including branding of MSME products. MSMEs also need to receive a briefing on the proper digital marketing strategy.

For B2C (Business to Customer) companies, a digital marketing strategy must focus on attracting potential customers to open the website and turning them into customers without having to communicate personally with the salesperson. To market products, MSMEs may adopt several marketing strategies, such as content marketing, mobile marketing, integrated digital marketing, continuous marketing, personalized marketing, and visual marketing. Understanding of how to determine the target market and stages in social media marketing, such as how to identify who the target consumers are, competitors, market segments, product analysis, and the importance of psychological understanding of target consumers to evaluate, are also needed.

Secondly, human resources development. Knowledge transfer can be focused on providing training to new employees; encouraging decision-makers in MSMEs to undertake professional education, such as knowledge related to selection and recruitment, reward, and punishment; as well as the role of leadership in business management. Thirdly, access to experts and facilities through developing and strengthening knowledge (both tacit and explicit) and skills, as well as utilizing university facilities to complement the available resources.

Universities might provide services in business development and incubators. For instance, Gadjah Mada University Yogyakarta has the Directorate of Business Development and Incubator that is responsible for managing and accelerating commercial processes. In terms of business development, this directorate is responsible for designing business development that supports teaching, research, and community service activities, coordinating economic activities between central and subsidiary businesses, facilitating incubation, and keeping businesses accountable in line with the university's duties and responsibilities. In terms of business incubation, the directorate is responsible for developing systems and business incubation models based on research conducted by universities and involving strategic partners.

Several studies on knowledge transfer from universities to MSMEs have been conducted (Ibidunni et al. 2020; Anand et al. 2021; Daat and Sanggenafa 2022). Ibidunni et al. (2020) conducted a survey of 370 MSMEs managers and owners in Nigeria to examine the effect of knowledge transfer on MSMEs' innovation performance. The results show that knowledge dimensions such as R&D and social networks have a significant impact on the innovation performance of MSMEs.

Anand et al. (2021) conducted a systematic literature review analysis to investigate the importance of knowledge transfer and sharing for MSMEs. The results of the study concluded that the transfer and sharing of knowledge were conducted to increase the strategic focus of MSMEs, especially regarding the aspects of human resources which include organizational learning and increased creativity. The results also concluded that the most important human factors related to the process of transferring and sharing knowledge to MSMEs include innovation, trust, and performance.

A study conducted by Daat and Sanggenafa (2022) tested the effect of knowledge-sharing practices on the performance of MSMEs mediated by human capital involving 47 MSMEs owners and management in Jayapura. The results of the study show that human capital does not mediate the effect of knowledge transfer on MSMEs' performance. The study also confirms the important role of knowledge-sharing practices in creating new knowledge, which plays an important role in determining the achievement of MSMEs' performance.

### 2.3. Hypotheses Development: Drivers and Barriers for MSMEs IR4 Readiness

The term IR4 was developed in 2011 in Germany when the German Federal Government initiated efforts to strengthen and improve the competitiveness of the country's manufacturing industry (Stentoft et al. 2019). IR4 is defined as a condition that involves technical integration with the Cyber–Physical System in manufacturing and logistics activities, as well as the use of the Internet of Things (IoT) in industrial processes, which will ultimately affect the value creation process, business model, downstream services, and how

the organization works. The existence of drivers and barriers is important to determine the level of organizational readiness to face IR4.

The term IR4 readiness is used to describe the ability of an organization to exploit and utilize digital technology in dealing with IR4. This level of readiness is distinguishable from the level of maturity. The level of readiness is assessed when an organization is not yet attached to the technology, while maturity is assessed after the organization implements the technology (Stentoft et al. 2019). Hanafiah et al. (2020) define IR4 readiness as the level of an organization's ability to take advantage of the implementation of IR4 technology. In other words, IR4 readiness shows the level of digitalization of an organization to enter the IR4 era. Assessment of organizational readiness to face IR4 is crucial to maintain the survival of the organization in the IR4 era. Organizations that are unable to adapt to and change with IR4 will face threats in maintaining organizational survival. This has attracted the interest of researchers to conduct studies on organizational readiness for IR4.

An example of a driving factor for IR4 readiness is human resources. To adapt to IR4, trained and skilled human resources related to digital technology in daily activities, such as robotics, nanotechnology, or astronautics, and able to communicate with robots with support from not only web technology but also intelligence support systems, are needed. An example of a barrier is the opposite: a low level of expertise and competence possessed by human resources (Turkes et al. 2019).

Turkes et al. (2019) conducted a study that aimed to identify the opinions and perceptions of managers on drivers and barriers in technology implementation to assess IR4 readiness. This study was conducted in Romania and involved 176 MSMEs managers who were asked questions through a closed-ended questionnaire. The results of the study concluded that MSMEs in Romania were in the stage of a full transition process from IR2 to IR4. Stentoft et al. (2019) stated that studies on organizational readiness in responding to the IR4 era have been mostly conducted in the manufacturing industry and are still very limited to MSMEs, causing research gaps on related issues.

Eltayeb et al. (2021) conducted a systematic literature review covering 22 articles and 1 textbook taken from Scopus, WoS, and Google Scholar to identify the drivers and barriers of IR4 and assist managers and practitioners to understand the key aspects of implementing IR4. Based on the results of the study, several drivers and barriers in the implementation of IR4 can be identified, including aspects such as economic, strategic, legislative, operational, process, culture and organization, human resources and competency, and environment and security.

For example, regarding the aspects of human resources and competence, improvement of technical and non-technical skills to adapt to changes in the business environment and technological developments has become one of the most important drivers of IR4 readiness. On the other hand, aspects of human resources and competence can also be a barrier to IR4 readiness if the following conditions exist: disruption in existing jobs, inadequate qualifications of the workforce, gaps in leadership skills, no concern from top management for IR4 when there are gaps in workforce skills, and a lack of digital skills and mastery.

Ghobakhloo et al. (2022) conducted a study using a systematic literature review approach to understand what factors influenced MSMEs in adopting IR4 technology in terms of technological, organizational, and environmental aspects. The results of the study found that MSMEs are still far behind in taking advantage of IR4 technology disruption opportunities when compared to large companies. MSMEs are still struggling with the decision to adopt technology to conduct digital-based business transformations in response to the demands of IR4. The researchers concluded that there are various technological, organizational, and environmental factors that determine the position of MSMEs in adapting to IR4. The main barrier factor for MSMEs is the complexity of IR4 technology; determinants such as the perceived benefits of IR4 or intrinsic pressure from stakeholders can be the main driving factor for MSMEs to digitize. However, there are several factors, such as digital technical competence or managerial competence, which might play a double role as a "double-edged sword" in facilitating the digitization of MSMEs.

Perera et al. (2023) conducted a study to examine the effect of drivers and barriers on digitalization involving 542 respondents. The survey results show that high levels of accuracy and trust, quality improvement and standards, the ability to complete complex projects with an accurate budget, time, and quality, and adequate communication between stakeholders are the four highest driving factors in the study. Accuracy and trust are the most important factors affecting performance and function in a construction company, as are quality, standards, delivery, and communication. With digitalization, MSMEs are expected to be able to correct deficiencies, eliminate errors, improve quality and standards, and improve collaboration between stakeholders, which ultimately affects the speed of the delivery process.

A study on the readiness of MSMEs to face IR4 in Indonesia was conducted by Sari and Santoso (2019) involving 100 respondents of younger-generation MSMEs in Karawang Regency. The results of the study show that the younger generation of MSMEs has been able to adapt to the use of digital technology and take advantage of the opportunities and market share resulting from their adoption of digital technology. However, these MSMEs still do not have a cooperation network, such as being involved in the MSMEs community, which is recognized as important as a channel to share experiences and information among MSME actors through coaching and mentoring.

Norman and Alamsjah (2020) conducted a study to identify the obstacles and key factors that determine the readiness of MSMEs in the textile and clothing (TC) sector in Indonesia by distributing questionnaires to TC sector owners and workers. The study results show that the five main inhibiting factors are high investment, lack of digital culture and training, lack of digital infrastructure, lack of government support and regulation, and ineffective change management. The researchers emphasize that to increase digitalization readiness, MSMEs are required not only to focus on minimizing or eliminating obstacles but also to strive to increase profit margins and form an innovative business ecosystem.

Tama et al. (2021) conducted a study on MSMEs in the automotive sector to determine their level of readiness using the IR4 readiness model created by VDMA. Data collection was carried out using survey methods and interviews with 57 MSMEs in the Automotive sectors in Indonesia. The results of the study found that most of the sample, namely 56%, were still at the initial adoption level, 14% had adopted at the outsider level and the remaining 30% still had no plans for digitizing. These findings indicate that the level of IR4 readiness for automotive MSMEs in Indonesia is still in the early adoption stage and requires support from the Government in terms of policies and regulations so that the digitalization process can be accelerated and the level of IR4 readiness can be increased.

The latest study on drivers and barriers to MSMEs' readiness in Indonesia to implement digitalization was conducted by Fridayani and Chiang (2022) on MSMEs in Yogyakarta. Data was collected through in-depth interviews with 50 MSMEs. The results of the study showed that MSMEs' decision to digitalize was motivated by internal factors due to the needs of MSMEs themselves, and external factors due to demands for digital transformation, which became drivers of the e-readiness of MSMEs. Based on the interview results, several factors inhibiting e-readiness were identified, such as skills to do business online, digital platforms, availability of internet infrastructure, and restrictions during the COVID-19 pandemic.

The hypothesis in this study was developed based on the results of a study conducted by Stentoft et al. (2019) that found that perceived drivers and barriers significantly influence MSME readiness in responding to IR4. Perceived drivers are identified as readiness or unreadiness in terms of legislation or laws and standards, strategies, manpower, and public advisor systems. By using a questionnaire survey method involving 308 MSMEs, the results of the study show that perceived drivers significantly affect the increasing MSMEs readiness, while barriers result in a decrease in MSMEs readiness levels; however, it has no significant impact on IR4 practices. The results of this study indicate the importance of planning the organizational transformation process toward the digitalization

process. Based on the results of these studies, the following hypotheses were developed in this study:

**H1.** *Perception of higher drivers for IR4 promotes IR4 readiness.*

**H2.** *Perception of higher barriers to IR4 decreases IR4 readiness.*

### 3. Methodology

This section discusses the methodology or research design that will be used in this study using a quantitative approach. Quantitative research methods are defined as part of a series of systematic investigations of phenomena by collecting data to then be measured by mathematical or computational statistical techniques. The discussion includes population, sampling methods, data collection techniques, variable measurement, validity and reliability testing, and data analysis techniques.

#### 3.1. Population, Sampling Method, and Technique of Data Collection

The research population was all MSMEs in Indonesia that met the criteria for assets and turnover in Law No. 20/2008. The sample was selected using the convenience sampling method. Convenience sampling is used based on the ease of use and easier for researchers to obtain data without considering several aspects, such as the use of randomized sampling. Other considerations are: Firstly, the limited space for researchers in the data collection process due to large-scale activity restrictions and lockdown policies that hindered the data collection process during the pandemic. Secondly, there are still many MSMEs that do not have email or have mastered digital technology, resulting in the filling process often requiring assistance from surveyors. This study used cross-sectional data, and data were collected through online surveys using Google Forms with owners as the respondent target.

#### 3.2. Variable Measurements

To find the IR4 readiness of MSMEs in Indonesia, this study modified the research of Stentoft et al. (2019) and Turkes et al. (2019), particularly to develop the variable measurements. The selection of these two studies was based on the researchers' findings that both were pioneers in the study of the effects of drivers and barriers to IR4 readiness on MSMEs based on the findings of the literature review conducted by the researchers.

The dependent variable was the measurement of MSME readiness for digital transformation, which can be defined as the degree to which MSMEs are able to exploit and derive benefits from IT technologies, known as MSME e-readiness. The dependent variable consisted of 7 statements, including pressure to work with IR4 from customers, suppliers, authorities, etc.; risks of experimenting with IR4; knowledge about IR4 to judge its importance for the company; support from top management to judge and work with IR4, competencies to work with IR4; motivation to judge and work with Industry 4.0; and economic freedom to work with Industry 4.0.

The independent variables consisted of the driver and the barrier factors. Driver factors refer to the supporting factors for MSMEs' e-readiness, while barrier factors refer to the inhibiting factors for MSMEs' e-readiness. The driver factors consisted of 10 statements, including customer requirements, competitors' practice, industry costs reduction, time-to-market improvement, legal requirements, lack of qualified workforce, competitor benchmark, public advisor initiative, consultant suggestion, and conscious strategy on IR4.

The barrier factors consisted of 11 statements, including lack of knowledge about IR4, lack of standards, more focus on operation at the expense of developing the company, lack of data protection, lack of employee readiness, requirement for continued education of employees, lack of understanding of the strategic importance of IR4, lack of understanding of the interplay between technology and human, too few financial resources, too few human resources, and data security uncertainties. All of the indicators used a 5 Likert Scale (1 = very low, 5 = very high).

Open questions were given to identify the problems faced by MSMEs in Indonesia and the knowledge transfer needed from universities to MSMEs. These two open questions were included, as based on the literature review conducted by the researchers, it was shown that MSMEs in Indonesia still have many problems, especially the internal problems related to some functional aspects, such as human resources, operations or production, marketing, and finance, as well as organizational management, which can be barriers for the digitalization of MSMEs. These internal problems are triggered by the limited resources owned by MSMEs and low digital literacy. To optimize the role of universities as knowledge-producing institutions in helping MSMEs to solve internal problems and increase digital literacy, we argue that knowledge transfer activities from universities to MSMEs are necessary.

### 3.3. Validity and Reliability Testing

Validity testing is carried out to determine the accuracy level of an instrument in carrying out its measuring function. Meanwhile, reliability testing is carried out to determine the consistency of an instrument; namely, whether the measuring instrument used is reliable and remains consistent when the measurement is repeated. Validity testing used Pearson Correlation and reliability testing used Cronbach's Alpha.

### 3.4. Technique of Data Analysis

Data were analyzed using descriptive statistics and multiple regression analysis. Descriptive statistics summarized the overall contents of the data, such as the mean (average data), standard deviations (how the data varies by group), and data variance. To test the hypotheses and to predict the effect of independent variables (X) on the dependent variable (Y), multiple regression analysis was used in this study. The open questions raised in this study were analyzed based on the answers or opinions given by the respondents.

## 4. Results

This section discusses data analysis, which includes a discussion of the respondent profiles and MSME business profiles involved in this study, validity and reliability testing results, descriptive statistics, hypothesis testing, and discussion of results.

### 4.1. Respondent and Business Profile

The questionnaire was sent to 350 potential target respondents and we assumed that there was no difference between the respondents who answered before and after the deadline for returning the questionnaire which was set. The online survey obtained 101 respondents who filled out the questionnaire completely and the rate of return yield was 28.86%. Table 1 summarizes the respondent and business profiles of the study.

Based on the data related to respondent characteristics and business profiles presented in Table 1, the following information was obtained: Most of the respondents who participated in this study were female, namely 55 respondents (54.46%), and the remaining 46 respondents (45.54%) were male. Most of the respondents (namely, 80) were owners (79.21%), in addition to 1 commissioner (0.99%), 7 managers (6.93%), and 13 staff (12.87%). The majority of MSMEs had been operating for 5–10 years (59.41%), 23.78% had operated for 10–29 years, 5.95% had operated for 20–30 years, and 10.88% had operated for more than 30 years.

Based on the type of business, the majority of MSMEs operate in the food and beverages sector (30.69%); 19.80% were in the wood, bamboo, rattan, handicraft, and furniture businesses; 14.85% were related to textiles, clothing, leather, and metal goods, machinery, automotive, electronics, and computers; 7.92% were in the chemical, oil, coal, rubber, and plastics industries, 5.95% were private education and consultant businesses, and 3.96% were logistics businesses.

**Table 1.** Respondent and business profiles.

| Dimension | Categories | Number of Respondents | Percentage |
|---|---|---|---|
| Gender | Male | 46 | 45.54 |
| | Female | 55 | 54.46 |
| Status | Owner | 80 | 79.21 |
| | Commissioner | 1 | 0.99 |
| | Manager | 7 | 6.93 |
| | Staff | 13 | 12.87 |
| Company age | 5–10 years | 60 | 59.41 |
| | 10–20 years | 24 | 23.76 |
| | 20–30 years | 6 | 5.95 |
| | More than 30 years | 11 | 10.88 |
| Type of business (Producing and/or marketing) | Textiles, clothing, leather | 15 | 14,85 |
| | Wood, bamboo, rattan, handicrafts, furniture | 20 | 19.80 |
| | Food and beverages | 31 | 30.69 |
| | Chemical, oil, coal, rubber, and plastics industries | 8 | 7.92 |
| | Non-metal goods, minerals | 2 | 1.98 |
| | Metal goods, machinery, automotives, electronics, and computers | 15 | 14.85 |
| | Private education and consultancy | 6 | 5.95 |
| | Logistics | 4 | 3.96 |
| Number of permanent workers | <5 workers | 66 | 65.35 |
| | 5–19 workers | 20 | 19.80 |
| | 2–99 workers | 8 | 7.92 |
| | >100 workers | 6 | 5.94 |
| | No answer | 1 | 0.9 |
| General performance within the past three years | Increase > 15% | 34 | 33.67 |
| | Increase 15% | 12 | 11.88 |
| | Increase < 15% | 10 | 9.90 |
| | No changes | 20 | 19.80 |
| | Decrease < 15% | 10 | 9.90 |
| | Decrease 15% | 5 | 4.95 |
| | Decrease > 15% | 10 | 9.90 |
| Assets | Less than or equal to IDR 50 million | 48 | 47.52 |
| | IDR >50 million–500 million | 35 | 34.65 |
| | IDR >500 million–10 billion | 18 | 17.83 |
| Turnover | Max IDR 300 million | 76 | 75.25 |
| | IDR >300 million–2.5 billion | 17 | 16.83 |
| | IDR >2.5 billion–50 billion | 8 | 7.92 |
| Collaboration with university | Yes | 10 | 9.91 |
| | No | 91 | 90.09 |

Source: Processed data.

Based on the number of employees, the majority had less than 5 employees (65.35%); 19.30% had 5–19 employees, 7.92% had 20–99 employees, 5.94% had more than 100 employees, and 0.9% did not answer. Based on the performance in the last 3 years, the majority of MSMEs answered that they had increased by more than 15% (33.67%), 19.80% claimed that their performance had not changed; 11.88% increased by 15%; 9.90% increased <15%, decreased <15%, and decreased >15%; and 4.95% experienced a 15% decrease in performance.

Based on assets, the majority of MSMEs stated that they had assets of less than IDR 50 million, namely 47.52%; 34.65% had assets between >50 million–500 million; and 17.83% had assets between IDR > 500 million–10 billion. Based on turnover, the majority of MSMEs stated they had a maximum turnover of IDR 300 million; 16.83% had a turnover of >IDR 300 million–2.5 billion; and 7.92% stated >2.5 billion–50 billion. Most of the respondents stated that they did not collaborate with universities (90.09%), while the remaining 9.9% stated that they had collaborated with universities.

### 4.2. Validity and Reliability Analysis

The validity testing result of all measurement indicators for the independent variable and dependent variables were valid based on r count value > r table value (0.195) with a significance level of 5%. Table 2 summarizes the validity and reliability testing for all variables tested in this study.

**Table 2.** Validity and reliability testing.

| Variable | Indicators | R Count | R Table 5% (100) | Sig. Value | Validity | Cronbach's Alpha |
|---|---|---|---|---|---|---|
| Drivers | D1 | 0.535 | 0.195 | 0.000 | Valid | 0.719 |
| | D2 | 0.558 | 0.195 | 0.000 | Valid | |
| | D3 | 0.504 | 0.195 | 0.000 | Valid | |
| | D4 | 0.501 | 0.195 | 0.000 | Valid | |
| | D5 | 0.440 | 0.195 | 0.000 | Valid | |
| | D6 | 0.569 | 0.195 | 0.000 | Valid | |
| | D7 | 0.664 | 0.195 | 0.000 | Valid | |
| | D8 | 0.590 | 0.195 | 0.000 | Valid | |
| | D9 | 0.511 | 0.195 | 0.000 | Valid | |
| | D10 | 0.516 | 0.195 | 0.000 | Valid | |
| Barriers | B1 | 0.668 | 0.195 | 0.000 | Valid | 0.857 |
| | B2 | 0.649 | 0.195 | 0.000 | Valid | |
| | B3 | 0.539 | 0.195 | 0.000 | Valid | |
| | B4 | 0.745 | 0.195 | 0.000 | Valid | |
| | B5 | 0.663 | 0.195 | 0.000 | Valid | |
| | B6 | 0.592 | 0.195 | 0.000 | Valid | |
| | B7 | 0.726 | 0.195 | 0.000 | Valid | |
| | B8 | 0.661 | 0.195 | 0.000 | Valid | |
| | B9 | 0.543 | 0.195 | 0.000 | Valid | |
| | B10 | 0.606 | 0.195 | 0.000 | Valid | |
| | B11 | 0.685 | 0.195 | 0.000 | Valid | |

**Table 2.** *Cont.*

| Variable | Indicators | R Count | R Table 5% (100) | Sig. Value | Validity | Cronbach's Alpha |
|---|---|---|---|---|---|---|
| | Red1 | 0.331 | 0.195 | 0.001 | Valid | |
| | Red2 | 0.578 | 0.195 | 0.000 | Valid | |
| | Red3 | 0.779 | 0.195 | 0.000 | Valid | |
| Readiness | Red4 | 0.739 | 0.195 | 0.000 | Valid | 0.775 |
| | Red5 | 0.791 | 0.195 | 0.000 | Valid | |
| | Red6 | 0.775 | 0.195 | 0.000 | Valid | |
| | Red7 | 0.637 | 0.195 | 0.000 | Valid | |

Source: Processed data.

The value of the r count for the driver's variable (0.440–0.664), the barrier variable (0.539–0.745), and the readiness variable (0.331–0.791). Cronbach's Alpha was used to test the reliability, and the instrument was considered to have high reliability if Cronbach's Alpha value was higher than 0.6 (Nunnally 1978). Cronbach's Alpha value ranged from 0.719–0.857, so it can be concluded that the measurements were reliable.

*4.3. Descriptive Statistic*

Table 3 summarizes the average respondents' answers to each indicator measured in the research variables, which include drivers, barriers, and IR4 readiness. Based on descriptive statistical analysis, it can be seen that the average readiness of MSMEs in Indonesia in responding to digital transformation ranged from 3.29–3.71, as well as related to drivers and barriers, ranging between (3.39–4.28) and (3.73–3.71). Based on the mean score of the respondents' answers, it was shown that they provide a neutral-to-high perception tendency for driver and barrier variables; however, e-readiness tended to be at a neutral value.

**Table 3.** Descriptive statistics.

| Variable | Mean | Std. Deviation |
|---|---|---|
| Drivers (D) | 3.39–4.28 | 0.739–1.040 |
| Barriers (B) | 3.73–4.11 | 0.868–1.076 |
| Readiness (R) | 3.29–3.71 | 0.896–1.176 |

Source: Processed data.

*4.4. Hypothesis Testing*

To test Hypothesis 1 and Hypothesis 2, this study uses multiple regression models. By testing the multiple linear regression model, the estimated parameter with the t value and the coefficient of determination ($R^2$) will be obtained. The regression coefficient is significant at $p < 0.05$, when it can be concluded that the independent variable significantly affects the dependent variable, and the greater the $R^2$, the better the model in explaining the variation in the dependent variable.

Table 4 summarizes the hypothesis testing results. All deviations from classical assumptions were tested. The normality assumption was tested by using a histogram, plot, and one-sample Kolmogorov–Smirnov. The histogram and plot tests show that the data were normally distributed, and the result of one-sample Kolmogorov–Smirnov shows the asymp. significant value is 0.200 or >0.05, meaning that the data were normally distributed. Multicollinearity was tested by the VIF value (1.694) or <10, and the tolerance value was 0.623 > 0.1, meaning there is no multicollinearity between the independent variables. Heteroscedasticity was tested by scatter plot residuals and showed that the points spread

randomly both above and below zero and the Y-axis so that it can be concluded that there is no heteroscedasticity.

**Table 4.** Hypothesis testing.

| Model | Stand. β | Stand. Error | t | Sig | F | Sig | Adj R$^2$ |
|---|---|---|---|---|---|---|---|
| Constant | 10.267 | 3.379 | 3.038 | 0.003 | | | |
| D | 0.442 | 1.06 | 4.171 | 0.000 | 11.167 | 0.000 | 0.169 |
| B | −0.060 | 0.075 | −0.805 | 0.423 | | | |

Source: Processed data.

The results of partial model testing show that the first hypothesis is supported, while the second hypothesis is not supported. This means that the first hypothesis, stating that perception of higher drivers for IR4 will promote higher IR4 readiness, is supported, with a significance value of 0.000 and the coefficient of standardized β 0.442 showing that the driver's variable has a positive influence on IR4 readiness. The second hypothesis, stating that the perception of higher barriers to IR4 will decrease or lower IR4 readiness, is not supported, with a significance value of 0.423, even though the coefficient of standardized β −0.040 shows that the barrier variable has a negative influence on IR4 readiness. The hypothesis testing shows that simultaneously, both perceptions of drivers and barriers will influence IR4 readiness. The adjusted R2 test is 0.169, meaning that 16.9% of MSME IR4 is explained by the variables of drivers and barriers, while 83.1% is explained by other variables outside this research model.

The first hypothesis, that the perception of higher drivers for IR4 promotes higher IR4 readiness, was supported and supports the previous finding of the study conducted by Stentoft et al. (2019). The second hypothesis, that the perception of higher barriers to IR4 decrease or lower IR4 readiness, was not supported. This finding contradicts the results of previous studies conducted by Stentoft et al. (2019) and might be explained by the perspective of the MSME's condition in Indonesia. Undeniably, the digital literacy level of MSMEs in Indonesia is still low.

These findings are in line with the studies conducted by Norman and Alamsjah (2020) and Tama et al. (2021), who conducted studies to identify the drivers and barriers of MSMEs' e-readiness in Indonesia. Norman and Alamsjah's study confirmed that the digital literacy of MSMEs in Indonesia is still low in the TC (Textile and Clothing) sector, which is caused by the limited financial resources of MSMEs; on the other hand, digitalization requires a high investment value. Another factor found in the study was low government support and regulation.

For example, in terms of access to financing and funding, there are many MSMEs in Indonesia that are still in the un-bankable category, so it becomes difficult to access financing and funding from formal financial institutions such as banks (Anatan and Ellitan 2023). Another factor is related to the low digital infrastructure in Indonesia; even though the Palapa Ring Project has been developed, there are still many MSMEs that do not receive adequate internet access not only because of the limited skills and competence of human resources, but also location factors, which may indeed be unable to receive adequate internet access (Adiningsih 2019). Other barrier factors are digital culture and training constraints, as well as resistance to change due to limited skills and competence of employees, as well as restrictions during the COVID-19 pandemic (Norman and Alamsjah 2020; Fridayani and Chiang 2022).

The finding of this study also confirms the study conducted by Tama et al. (2021), which found that in the automotive sector, the industry's IR4 readiness level is still low. The digitalization of MSMEs is still in the early stages of adoption; 30% of MSMEs in the study even stated that they did not have an obvious plan for digitizing. This indicates that the level of digital literacy of MSMEs in Indonesia is still low, so the effect of barriers on the e-readiness of MSMEs is not significant.

To discover the internal problems faced by MSMEs, in this study, respondents were asked open questions regarding the problems they faced in the aspects of finance, operations/production, marketing, human resources, and company management. Through identifying problems in each functional area, it is hoped that MSMEs can understand what knowledge requirements are needed but not owned by companies, and can be accessed from outside the company; for instance, from universities through knowledge transfer activities from universities to MSMEs.

Based on the respondents' responses regarding the problems faced by MSMEs, they can be classified into financial, operational, marketing, human resources, and organizational management problems. Based on the financial aspect, problems faced by MSMEs are related to the limited access to funding; the cost of SWAB, PCR, and quarantine during the pandemic, leading to difficulties in MSMEs operating normally; limited working capital; difficulties paying capital loans; financial records still being limited and simple, which focus on the cash flow; high material costs not being worth the income; high investment to implement digital technology as business capital; budgeting for MSMEs not being conducted properly; and no separation between personal and business assets.

Based on the operational aspect, the problems faced by Indonesian MSMEs can be identified as follows: constricted product price competition; inadequate inventory and quality control systems; lack of knowledge related to the aspects of production and business operation; the employee skill of production being low, leading to low productivity; no obvious production schedule; high storage cost; lack of support from other parties, such as fund providers, suppliers, and distributors; lack of innovation, meaning that MSMEs cannot compete with other parties that operate in a similar industry; difficulties in obtaining raw materials on time; limited and inadequate production equipment; MSMEs operating hours being limited due to the pandemic; and no Standard Operating Procedures (SOP).

From the marketing aspect, the problems faced by Indonesian MSMEs can be identified as follows: limited market share; promotion being less reliant on word of mouth, and marketing only relies on social media, such as Instagram, Facebook, and WhatsApp; fewer networks; limited distribution channels; products marketed being homogeneous; lack of knowledge and the use of digital marketing; MSMEs usually do not have marketing staff; little proactivity in marketing products, without brand and packaging; and lack of cooperation with other parties.

Based on human resource aspects, problems faced by the Indonesian MSMEs can be identified as follows: low skill and competence of employees; less able to cooperate; lack of knowledge related to management and how to manage a division within the company; lack of technology; lack of motivation and initiative to learn new things; resistance to change; lack of sense of belonging to the workplace; lack of training and development activities; low trust in fellow employees; and difficulty in the succession process.

Based on the administrative and business management aspects, problems faced by the Indonesian MSMEs can be identified as follows: administrative records still being managed by the owner; lack of knowledge related to company administration; paperless administrative work still rarely being implemented by MSMEs; MSMEs have not adopted and utilized information and communication technology optimally; and administration activities are still chaotic and unorganized. Related to business management, traditional and conventional management is usually conducted where the business is managed by the owner and family, the implementation of group-based management, the decision-making depends on the manager, and business management is still the owner's responsibility.

Respondents were also asked to identify the needs for knowledge that might be delivered in training and development from universities to MSMEs. Regarding the knowledge transfer activities needed by MSMEs from universities to expand their skill and business, the types of knowledge transfer can be identified as follows: business management; information and communication technology implementation; company branding; financial management for MSMEs; competitive strategy in IR4 era; a strategy to expand market share; entrepreneurship and leadership in doing business; human resource empowerment;

online business strategy and the role of influencer; digital marketing and content strategy; PLC Programming; and integration, automatization, and digital business.

MSMEs' knowledge needs to be increased to succeed in the MSME digitization process. Knowledge might be easily accessed through both internal and external knowledge transfer activities, such as through collaboration with universities as the producers of knowledge. Knowledge transfer activities from universities to MSMEs or external knowledge are expected to overcome the limited knowledge and competencies possessed by MSMEs. External knowledge is important in improving performance and competitiveness and might be obtained from various sources, such as consumers, universities, experts in the field of intellectual property rights, and partners (Brunswicker and Vanhaverbeke 2014).

Previous studies on knowledge transfer activities focused more on large-scale companies, while MSMEs, especially in developing countries such as Indonesia, still tended to be low and slow due to internal factors (such as organizational conditions and management characteristics) and external factors (such as the rapid development of technology and organizational environment). The use of information and communication technology (ICT) in MSMEs, such as computers, internet access, and communication, for finding information, placing orders, receiving payments, customer service, and purchases and payments to vendors, tends to be limited.

Ramdansyah and Taufik (2017) argued that along with the increase in the use of ICT, the use of the Internet and the adoption of electronic commerce (e-commerce) in Indonesia has also increased. However, only a few MSMEs in Indonesia have taken advantage of ICT developments and adopted e-commerce. On the other hand, the rapid development of ICT requires MSMEs to adopt ICT to survive and succeed in the increasingly fierce competition. Knowledge transfer from universities as a source of innovation in MSMEs is believed to be an important factor in supporting the survival and e-readiness of MSMEs (Braun and Hadwiger 2011). Previously conducted studies have confirmed the important role of knowledge transfer activities from universities to MSMEs, not only to optimize the university's role in managing and empowering MSMEs, but also in finding solutions to overcome problems and increasing the readiness of MSMEs to carry out digital-based business transformations (Ibidunni et al. 2020; Anand et al. 2021; Daat and Sanggenafa 2022).

## 5. Discussion

In the hypothesis testing section, a comparison of the findings in this study with previous studies has been discussed. Based on the results, it can be concluded that the findings in this study have contributed to broadening knowledge regarding the MSMEs' e-readiness testing model. This can be explained by several reasons: First, so far studies on e-readiness have been mostly applied to large-scale industries; however, due to IR4 and the COVID-19 pandemic, MSMEs must implement digitalization so that they are able to survive in conditions where many restrictions are imposed to minimize physical contact between individuals and goods.

Second, studies on IR4 drivers and barriers in Indonesia so far have focused on only one industry, such as a study conducted by Norman and Alamsjah (2020), which focused on the textile and clothing industry, and Tama et al. (2021), who focused on the automotive industry. In this study, several types of industries were involved; however, this study did not control the industrial effects. Third, previous studies have focused on industry per region; for example, Putri and Asyari (2023) conducted a study for MSMEs in Bukittinggi, Indriastuti and Kartika (2022) focused on studies for the Central Java region. In this study, the research was focused on three provinces in Java Island, including West Java, Central Java, and East Java.

Fourth, previous studies on drivers and barriers to IR4 in Indonesia focused more on identifying the drivers and barriers faced by MSMEs when they decided to go digital. In this study, the research focus was to confirm the findings of the study of Stentoft et al. (2019), which was carried out through hypothesis testing. This study also identified

internal problems faced by MSMEs in preparing for digitization and identified the need for knowledge transfer based on MSMEs' perceptions of internal problems related to the functional aspects of the company.

The fourth previous discussion on how this study expands the knowledge also explains the novelty of this study, considering that studies on the drivers and barriers to MSMEs' e-readiness in Indonesia so far have focused on identifying the drivers and barriers of MSMEs' e-readiness.

This study's findings are considered to be economically significant, considering that it has been empirically proven that digitization will have a significant effect specifically on the financial performance of MSMEs (Indriastuti and Kartika 2022; Gunawan and Somantri 2023), MSMEs' revitalization (Putri and Asyari 2023), and in general on the national economy (Nata et al. 2022). The level of e-readiness of MSMEs determines the success of the digitization process, which will provide advantages for MSMEs, especially from an economic point of view. Digital transformation allows the automation of operational activities in business, which has an impact on operational efficiency, such as by achieving a reduction in transaction costs, which will ultimately affect productivity.

The Minister of Cooperatives and Small and Medium Enterprises, Teten Masduki, stated that in 2021, of the total population of MSMEs in Indonesia, only around 12 million or 19% of MSMEs utilized digital platforms and were still faced with various obstacles in the digitization process. Assessment of digital literacy levels based on the ability to operate digital devices, applications, and platforms is required to support the effectiveness of using digital technology for MSMEs (Natalia 2021).

To support the increase in digital literacy of MSMEs in Indonesia, recently, the Ministry of Cooperatives and MSMEs involved MicroSave Consulting by optimizing the access and capacity of cooperatives and MSMEs. Through this collaboration, it is expected that the Ministry of Cooperatives and SMEs might formulate policies and program plans which support efforts to increase MSMEs' digital literacy (Catriana 2021). Another collaboration might be carried out by involving universities as knowledge producers through the mechanism of knowledge transfer activities from universities to MSMEs as an alternative strategy to overcome the problems and increase MSMEs' e-readiness for IR4. An effective model of knowledge transfer from universities to MSMEs needs to be developed. Chonsawat and Sopadang (2020) stated that the external knowledge obtained by MSMEs from universities will determine the level of readiness of MSMEs in adopting technology and determine the success or failure of MSMEs to transform into MSME 4.0.

## 6. Conclusions

The results of the research model testing show that the first hypothesis, that perceptions of higher drivers for IR4 promote higher IR4 readiness, was accepted, while the second hypothesis, that perceptions of higher barriers for IR4 decrease IR4 readiness, was not supported. The main problems faced by MSMEs are related to financial, human resources, marketing, operational, administrative, and organizational management. Based on open-ended questions posed to respondents, several topics of knowledge transfer from universities required by MSMEs include business management, ICT implementation in business, branding, financial management, digital marketing and advertising, PLC programming, human resource management, and business strategy.

The managerial implications of this study finding can be identified as follows: First, to increase the level of readiness for the digitalization of MSMEs, a solid support is needed from stakeholders, such as the Governments (both Central and Regional) through obvious regulations on digitizing MSMEs and various support programs for improving the digital literacy of MSMEs. Financial Institutions also have an important role in providing access to funding and financing for MSMEs to be able to carry out digital transformation.

In addition, educational and training institutions such as universities, as knowledge producers, need to optimize their role to solve the internal problems of MSMEs. For example, the problem of limited knowledge and competence in human resources can be

overcome by providing the transfer of knowledge related to existing problems, such as the introduction of recording cash flows; debt management related to financial aspects; the introduction of digital marketing, branding, and customer relationships; management related to marketing aspects, or an introduction to quality management; demand forecasting; and production scheduling related to operational aspects.

Second, considering the importance of drivers and barriers in encouraging or hindering e-readiness levels, MSMEs and related stakeholders need to focus on managing both. For example, when the aspect of human resources plays an important role in increasing the level of readiness, then the provisions related to digital skills need to be increased; or, when the funding problem is the biggest barrier, then a solution to find funding needs to be obtained immediately.

Third, there are still many internal problems related to aspects of human resources, operations, marketing, finance, and business management. Solutions need to be immediately sought, one of which is to gain access to knowledge from external parties such as universities. Strategic partnerships between universities and MSMEs need to be developed so that knowledge transfer activities from universities to MSMEs can be conducted, and universities, as knowledge producers, can optimize their role to overcome internal problems faced by MSMEs due to limited resources and low digital literacy.

The main limitation of this study lies in the sample size and data collection process. Considering that data collection was carried out during the period of large-scale social restrictions policy implementation, namely April–August 2021, the researchers faced difficulties in collecting data since they could only rely on online surveys via Google Forms, whereas in fact there are still many micro-scale MSMEs who have not mastered the technology well enough to access this platform. Therefore, the sample size in this study was not determined and the data were processed according to the return of a filled questionnaire.

Data were collected only through an online survey due to some restrictions during the pandemic; on the other hand, not all MSMEs are technology-literate or may understand the research questions given, so researchers need to guide and provide some explanations while the target respondents are filling out questionnaires. The difficulty of obtaining these data and the achievements of the processed data are unoptimized and might lead to unoptimized results as well. The results of the study might not be able to represent the overall character of the MSME population in Indonesia, since MSMEs who do not master technology will automatically be excluded as potential respondents, except for MSMEs who request directions from researchers on how to fill out the Google form as explained in the Methods section.

Regarding suggestions for future research, this study is expected to reach a wider range of MSMEs operating in different types of businesses and include other variables that may affect the readiness of MSMEs in responding to the IR4 era, such as collaboration, knowledge transfer activities, the role of the Government, and other factors such as competitors or business environment. Future research agendas might focus on mapping the results of previous research on the issue of knowledge transfer from universities to MSMEs to develop a model of the effectiveness of knowledge transfer from universities to MSMEs. Control variables such as age, size, and type of business in future research can be examined and analyzed to investigate the role of control variables in minimizing or eliminating the influence of other variables aside from the independent variables (drivers and barriers) on the dependent variable (e-readiness).

This research is expected to provide insight and knowledge for academics, practitioners, and the public regarding the readiness of MSMEs in carrying out digital-based business transformation by considering the drivers and barriers of IR4 readiness. For academics, the results of the study are expected to enrich the literature on strategic management and entrepreneurship, especially regarding MSMEs digitization strategies, inter-organizational partnership strategies, and knowledge management. For practitioners, the results of the study are expected to provide insight as material for consideration in making decisions related to the issues studied, both digitizing MSMEs, solving MSMEs' problems through

knowledge transfer from universities to industry, as well as the types of knowledge needed by MSMEs to overcome the problem of limited resources, especially knowledge and low digital literacy. For the community, it is hoped that the results of this study will provide additional knowledge about the issues under study.

**Author Contributions:** Conceptualization, L.A.; methodology, L.A.; software, L.A.; validation, L.A.; formal analysis, L.A.; investigation, L.A. and N.; resources, L.A.; data curation, L.A. and N.; writing—original draft preparation, L.A.; writing—review and editing, L.A. All authors have read and agreed to the published version of the manuscript.

**Funding:** This research was funded by Badan Riset dan Inovasi Nasional (BRIN), grant number: 1867/E4/AK.04/2021; 020/SP2H/RDPKR-MONO/LL4/2021; 235-B/LPPM/UKM/VII/2021.

**Informed Consent Statement:** Not applicable.

**Data Availability Statement:** Not applicable.

**Acknowledgments:** We gratefully appreciate The National Research and Innovation Agency, Republic of Indonesia for the research grant provided through Penelitian Dasar Unggulan Perguruan Tinggi (PDUPT) Scheme. We also would like to thank Yayasan Perguruan Tinggi Kristen Maranatha (YP-TKM) and Lembaga Penelitian dan Pengabdian Masyarakat (LPPM) Maranatha Christian University for the support.

**Conflicts of Interest:** The authors declare no conflict of interest.

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
