# Peer review of "Micro, Small, and Medium Enterprises’ Readiness for Digital Transformation in Indonesia"

_economies, doi:10.3390/economies11060156_

Round 1
Reviewer 1 Report
- Transition paragrapghs must be added between sections no. 2. and 2.1., 3. and 3.1, and so on...
- Discussions must be improved with additional literature to formulate comparisons between the results and the existing literature.
- Conclusions must present the scientific novelty in a better way.
Wish you good luck!
Reviewer 2 Report
General Overview:
Even though the study carried out in the present research focuses specifically on Indonesia, the topic addressed is an extremely important one in the current context, an aspect to be appreciated.
The title of the article attracts the reader’s attention, while also comprising key terms that could be easily found during an intelligent search.
The authors managed to combine qualitative and quantitative research, a fact that adds value to the results obtained. To a large extent, the research follows a natural course, while the results are presented in an appropriate manner.
Recommendations:
- Since generalizing can be considered inadequate, it is recommended to mention or indicate in the title of the article that the research is related to the situation encountered in Indonesia.
- In order to highlight both the interest in the research topic addressed, as well as the efforts of the authors to document themselves on the subject, the need to deepen the specialized literature is felt, especially considering the review of more recent studies.
- The presentation of the results can be improved and extended. In certain cases, simply placing the information obtained in tabular form is not enough. There is a possibility that the audience can expect some perceptions/views of the authors regarding the data obtained. For example, even within the section 4.1 Respondent and Business Profile, although it may seem sufficiently explicit, the results obtained could have various meanings to discuss.
- Similarly, the Discussion section could benefit from improvements.
- Several discrepancies were observed between the present paper and required template of the journal. Therefore, it would be advisable for the author/s to review the structure and the design of the specific parts within the manuscript.
Reviewer 3 Report
The study deals with the current and significant topic.
However, it has the following shortcomings:
The lack of specific details on the methodology used in the study and the sample size.
Odd approach for data collection process, which affected the representativeness and generalizability of the results.
Open-ended questions posed to respondents may not have been structured enough to elicit specific responses.
The use of the online survey may have excluded MSME actors who are not technology-literate.
In the line (262 - 264) is written: "To find the IR4 readiness of MSME, this study modified research of Stentoft et al. (2019) 262 and Turkes et al. (2019). The dependent variable measurement of MSME readiness con- 263 sists of 7 items of statements, while the independent variables consist of 10 items for the 264 driver’s factor and 11 items of statements for the barriers factor"
Specific variables and specific reasons for changing the research results from Turkes and Stentoft are not provided.
The article does not state how the findings (confirmed hypothesis) extend scientific knowledge.
The theoretical part contains a small amount of scientific sources and cannot be considered a solid basis for further research.
The level of English language needs to be improved.
Reviewer 4 Report
The analytical review is very weak, it does not take into account publications in the world's leading journals. In addition, few previously performed studies on improving the activities of entrepreneurs using digital technologies have been considered. The hypotheses put forward are not substantiated. There are no economically significant results. The essence of digitalization in small and medium-sized enterprises is not disclosed. The goals and objectives of the study are not clear. Conclusions of interest to readers have not been formulated.
Round 2
Reviewer 2 Report
-
Author Response
Please see the attachment.
Regards,
Author

Reviewer 3 Report
The authors have significantly added to and improved their article. Therefore, the article can be accepted.
Recommendation:
The level of English language (especially in the parts that have not been changed) would benefit from improvement.
Author Response

(The authors gave the same response as above.)

Reviewer 4 Report
Read text to make the appropriate form for the quality of the journal, you can see some minor errors in the words.
